# Mechanism and inhibition of *Streptococcus pneumoniae* IgA1 protease

Zhiming Wang[1,5], Jeremy Rahkola[2,5], Jasmina S. Redzic[1], Ying-Chih Chi[3], Norman Tran[4], Todd Holyoak [4], Hongjin Zheng [1✉], Edward Janoff [2✉] & Elan Eisenmesser [1✉]

Opportunistic pathogens such as *Streptococcus pneumoniae* secrete a giant metalloprotease virulence factor responsible for cleaving host IgA1, yet the molecular mechanism has remained unknown since their discovery nearly 30 years ago despite the potential for developing vaccines that target these enzymes to block infection. Here we show through a series of cryo-electron microscopy single particle reconstructions how the *Streptococcus pneumoniae* IgA1 protease facilitates IgA1 substrate recognition and how this can be inhibited. Specifically, the *Streptococcus pneumoniae* IgA1 protease subscribes to an active-site-gated mechanism where a domain undergoes a 10.0 Å movement to facilitate cleavage. Monoclonal antibody binding inhibits this conformational change, providing a direct means to block infection at the host interface. These structural studies explain decades of biological and biochemical studies and provides a general strategy to block *Streptococcus pneumoniae* IgA1 protease activity to potentially prevent infection.

[1] Department of Biochemistry and Molecular Genetics, School of Medicine, University of Colorado Denver, School of Medicine, Aurora, CO 80045, USA. [2] Mucosal and Vaccine Research Program Colorado, Division of Infectious Disease, University of Colorado Denver School of Medicine and Denver Veterans Affairs Medical Center, Aurora, CO 80045, USA. [3] Cryo-EM Center, Vagelos College of Physicians and Surgeons, Columbia University Irving Medical Center, New York, NY 10032, USA. [4] Department of Biology, University of Waterloo, Waterloo, ON, Canada N2L 3G1. [5] These authors contributed equally: Zhiming Wang, Jeremy Rahkola. ✉email: Hongjin.Zhing@ucdenver.edu; Edward.Janoff@cuanschutz.edu; Elan.Eisenmesser@ucdenver.edu

Streptococcus pneumoniae (S. pneumoniae) is a leading cause of bacterial pneumonia and meningitis in children and adults worldwide[1–4]. According to the World Health Organization (WHO), S. pneumoniae remains a "major global public health problem". Polysaccharide-based vaccines have proven effective against invasive S. pneumoniae, but less so against mucosal infections, and these vaccines target only a subset of the known serotypes[5]. Thus, employing a more widely shared protein virulence factor such as the S. pneumoniae IgA1 protease (IgA1P) as a vaccine target has been advocated, as it is present in all pathogenic strains and active at the respiratory mucosa[6–11].

S. pneumoniae IgA1P is the prototypical member of the M26 class of bacterial metalloproteases[12], which share virtually no sequence homology to previously characterized proteins[9]. Other opportunistic pathogens that secrete similar IgA1 metalloproteases include Streptococcus oralis and Streptococcus sanguinis[9]. The mature forms of these enzymes comprise nearly 2000 amino acids and are covalently linked to their bacterial cell surface by sortase A[13]. Beyond the metalloprotease class of IgA1Ps, there are two other structurally distinct IgA1P classes that include serine proteases[14] and cysteine proteases[15], illustrating the evolutionary need to converge upon mechanisms of IgA1 cleavage in order to thwart the initial host immune response. All classes of bacterial IgA1Ps cleave host IgA1 within the heavy-chain (HC) hinge region, the linker connecting the IgA1 constant fragment (Fc) to its antigen binding region (Fab) (Fig. 1a), leaving the light chain (LC) intact while separating the Fc and Fab. The shorter linker within IgA2 and IgGs renders them inaccessible to IgA1P cleavage[16]. Proteolysis by IgA1Ps serves two purposes: to prevent phagocyte killing by decoupling the IgA1 Fc recognized by neutrophils from the pathogen-recognizing Fab[6,17] and to coat the bacterial surface with non-functional Fab fragments to effectively shield it from immune surveillance[18,19]. Despite the discovery of the metalloprotease class of bacterial virulence factors 30 years ago[20], their structure and mechanism of substrate engagement has remained unknown until now.

In this work, we use cryo-electron microscopy (cryo-EM) to elucidate the structure of the S. pneumoniae IgA1P catalytic region alone and in complex with both its IgA1 substrate and a neutralizing monoclonal antibody (mAb), thereby addressing the molecular basis of substrate recognition and enzyme inhibition.

## Results

### The high-resolution structure of the IgA1P catalytic region.
In order to identify the S. pneumoniae IgA1P catalytic domain, we engineered several constructs based on our previous results of limited proteolysis on the full, mature IgA1P (residues 154–1963, UniProt accession Q59947; NCBI accession WP_000417171 that corresponds to the common D39 and R6 strains)[21,22]. Only constructs that began at or prior to residue 665 were accessible to enzymatic cleavage of the MBP tag by thrombin (Supplementary Fig. 1a) and there was an observed reduction in IgA1-cleavage for shorter constructs (Supplementary Fig. 1b). Thus, we focused our cryo-EM studies on a construct of IgA1P spainning residues 665–1963 that could be excised from its MBP tag and had comparable cleavage to the full-length IgA1P (Fig. 1b).

The 3D cryo-EM reconstruction of the S. pneumoniae IgA1P (residues 665–1963) resulted in a 3.8 Å resolution map (resolution shells are shown in Fig. 1c and structural data parameters and refinement statistics are presented in Supplementary Fig. 2 and Supplementary Table 1). The overall structure reveals that S. pneumoniae IgA1P is a multi-domain enzyme (Fig. 1d), broadly comprised of N-terminal (NTD; residues 665–1070), middle (MD; 1071–1611), and C-terminal domains (CTD; 1612–1963). The NTD can be further divided into a small subdomain

(residues 665–769) attached by a long linker to a β-helix (residues 781–1070), a common structural motif found in several bacterial proteins[23]. A defining feature of this S. pneumoniae β-helix is both its relatively small size and the lack of protruding secondary structure elements from within the domain itself. For example, the β-helix of the functionally similar but structurally unrelated serine IgA1P from Haemophilus influenzae is about twice as long and has multiple domains that emanate from and return to the β-helical core[24]. The IgA1P MD and CTD have no structural similarity to any known protein to date, as indicated by the lack of structural similarity using DALI searches. The active site is formed between the MD and CTD domains that bifurcate the Zn-coordinating residues of the HEMTH motif (residues 1604–1608 in the MD) and a downstream E (E1628 in the CTD) (Fig. 1d, e). Despite the unique structural folds of the IgA1P MD and CTD, the S. pneumoniae IgA1P active site comprises catalytically important residues within similar positions to that of the prototypical metalloprotease, thermolysin (Fig. 1f), which supports a conserved catalytic mechanism that is consistent with our previous biochemical studies[21]. Specifically, three side chains from IgA1P H1604, H1608, and the downstream E1628 coordinate the Zn atom and are homologous with thermolysin residues H142, H146, and E166, respectively. The Zn both polarizes the carbonyl group of the scissile peptide bond and activates the nucleophilic water molecule that is deprotonated by a fourth residue, E1605 (E143 in thermolysin)[25].

### Trapping an active IgA1P/IgA1 complex.
In order to trap the S. pneumoniae IgA1P in complex with its IgA1 substrate, several challenges were overcome. We engineered an active-site mutant in the context of IgA1P 665–1963 that removes one of the Zn-coordinating residues, referred to as IgA1P-E1605A. While this mutation abrogates catalysis under catalytic concentrations of the enzyme (Supplementary Fig. 1b), catalysis was still observed during incubation at stoichiometric concentrations of the IgA1P-E1605A/IgA1 complex (Supplementary Fig. 1c). As EDTA was found to slow catalysis (Supplementary Fig. 1c), the IgA1P-E1605A/IgA1 complex was purified for cryo-EM studies in the presence of EDTA (Supplementary Fig. 1d). A cryo-EM 3D reconstruction of the complex resulted in a 3.5 Å resolution map (Resolution shells are shown in Fig. 2a and structural data parameters and refinement statistics in Supplementary Fig. 2 and Supplementary Table 1), which facilitated fits of the individual domains with subsequent refinement.

### The high-resolution structure of the IgA1P/IgA1 complex.
The structure of the IgA1P-E1605A/IgA1 complex reveals key details regarding their interaction that includes a gating mechanism that is described further below. Regarding the complex structure, the IgA1P NTD binds to the substrate IgA1 Fab (of the HC) on one end while the IgA1P CTD engages both IgA1 Fc monomers at the other end (Fig. 2b, c). Thus, the stoichiometry of a 1:1 complex can now be understood by the fact that a single IgA1P requires interactions with both IgA1 Fc monomers and therefore prevents two IgA1Ps from symmetrically binding two substrate IgA1s. In fact, weak density can be ascribed to the unbound Fab (Fig. 2a), supporting the fact that despite two IgA1 hinge sites may be available, only one is accessed at a time by IgA1P. The ability of IgA1P to broadly cleave polyclonal IgA1 is also explained by this structure, as IgA1P does not interact with the variable region of the IgA1 substrate (Supplementary Fig. 3).

As the highest resolution information on the complex was found within the active site, most of the IgA1 hinge backbone of the IgA1 hinge could be confidently modeled (Fig. 2d, e and Supplementary Fig. 2d). Following the numbering scheme of the

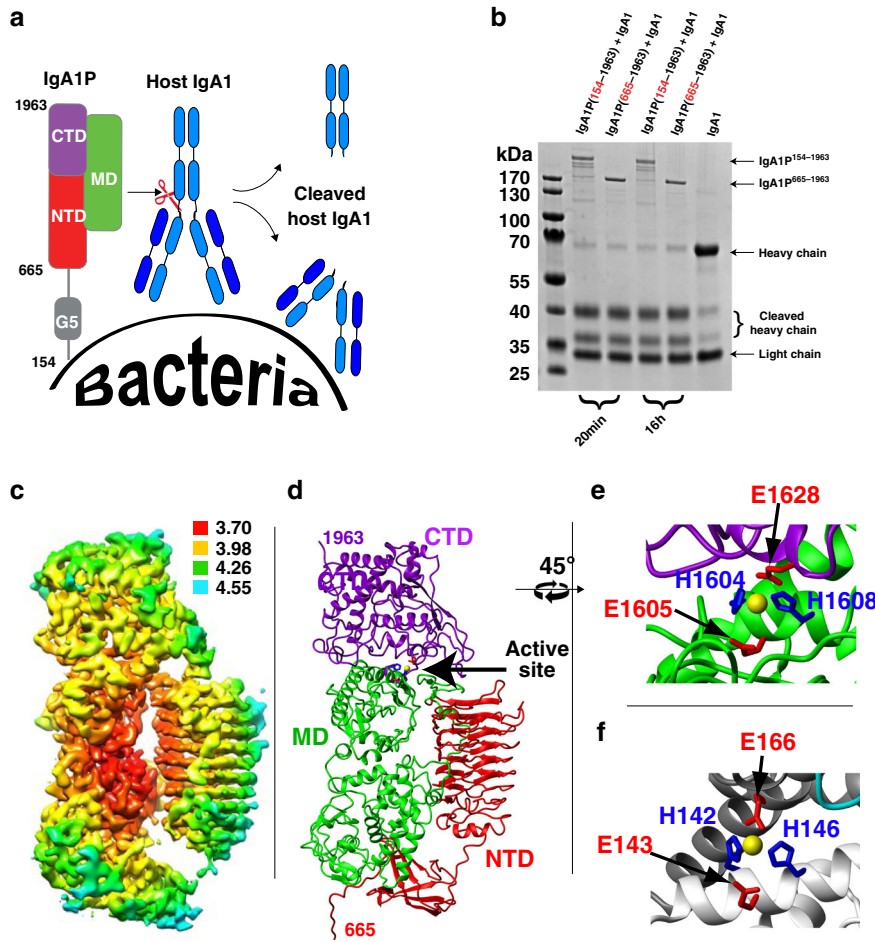

**Fig. 1 Function and cryo-EM structure of the *S. pneumoniae* IgA1P. a** The family of IgA1Ps cleave host IgA1 at its hinge region, separating the IgA1 Fc from its Fab and effectively masking bacterial cells with host IgA1 Fab. The general domain architecture of mature *S. pneumoniae* IgA1P consists of a flexible N-terminal region (residues 154–664) attached to the bacterial cell wall, which includes a small G5 domain followed by a large C-terminal catalytic region (residues 665–1963)[21,22]. **b** *S. pneumoniae* IgA1P cleavage is similar between the mature IgA1P 154–1963 and isolated catalytic region of IgA1P 665–1963. Shown is one of two independent SDS-PAGE gel measurements. **c** 3D reconstruction of *S. pneumoniae* IgA1P (residues 665–1963) is colored according to the local resolution estimates (units in Å). The map was produced using Chimera[38]. **d** *S. pneumoniae* IgA1P ribbon structure highlights the tertiary structure (PDB ID: 6XJB). Domains are each color-coded along with the modeled Zn ion placed based on the superposition of catalytic residues with those found in thermolysin (yellow, arrow). The structure was modeled using Coot[32]. **e** Expansion of the *S. pneumoniae* IgA1P active site with H1604, E1605, H1608, and E1628 shown. **f** Expansion of the thermolysin active site in a similar orientation to panel E (PDB ID: 1TLX). Functionally homologous thermolysin and IgA1P catalytic residues have the same color coding between panels **e** and **f**.

initial fit IgA1 Fab and Fc models[26,27], the HC comprising residues 1–221 within the initial Fab model could be extended through the active site of IgA1P to residue S232 (Fig. 2d). Although there is no visible density from residues 233–240, which connects the remaining hinge within the IgA1 HC, it is clear that IgA1 residues P227 and T228 are properly aligned for cleavage within the IgA1P cleavage site consistent with previous biochemical studies[28]. Thus, we were able to capture the intact active site within the active complex.

**The IgA1P gating mechanism.** Potentially the most striking finding of the *S. pneumoniae* IgA1P/IgA1 complex is the conformational change associated with substrate binding (Fig. 3), which underlies an active-site gating mechanism. This large-scale movement is best illustrated by a superposition of the *S. pneumoniae* IgA1P free and bound states (Fig. 3a), which results in a 7.2 Å RMSD for the whole enzyme but only a 1.5 Å RMSD when the NTD β-helix is excluded from the RMSD calculation. Thus, it is the NTD that repositions from a "closed" state in the absence of substrate to an "open" state allowing for binding of the IgA1

hinge region. In order to facilitate this domain rearrangement, two flexible loops are involved: IgA1P residues 770–783 and a 66-residue linker that connects the IgA1P NTD to the MD (residues 1051–1116). These loops act as flexible tethers to the entire β-helix (779–1050), providing enough slack to allow for a shift of the β-helix by approximately 10 Å relative to the rest of the protease (Fig. 3b).

The number of interactions between the IgA1P NTD and MD are diminished upon this rearrangement, providing for an energetic rationale for why the enzyme "snaps" back after product release. Such findings also led us to hypothesize that the IgA1P NTD may be soluble alone, similar to its G5 domain and CTD that have been previously shown to be independently folded[21,22]. Indeed, we were able to engineer an NTD construct that gives rise to a well-dispersed ¹⁵N-HSQC analogous to the independently folded IgA1P CTD and the G5 domain (Supplementary Fig. 4), supporting our findings that the NTD is capable of decoupling from the MD during substrate binding. Thus, *S. pneumoniae* IgA1P exhibits an active-site-gating mechanism in which the NTD β-helix plays a crucial role. Interestingly, a gating

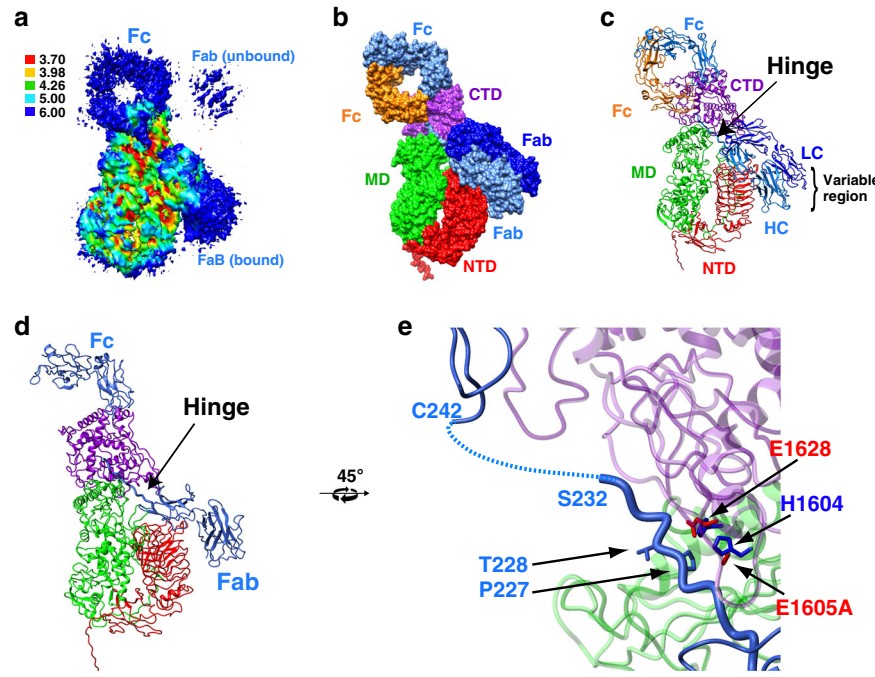

**Fig. 2 Cryo-EM structure of the *S. pneumoniae* IgA1P/IgA1 complex. a** 3D reconstruction of *S. pneumoniae* IgA1P-E1605A in complex with IgA1 is colored according to local resolution estimates (units in Å). The map was produced using Chimera[38]. **b** Surface representation of the IgA1P/IgA1 complex (PDB ID: 6XJA). The model comprises IgA1P residues 665–1963 (domains colored as in Fig. 1d), the IgA1 LC (dark blue), the IgA1 HC portion of the Fab (light blue), and the IgA1 Fc (orange and light blue for each HC monomer). **c** Ribbon structure of the IgA1P/IgA1 complex. The hinge connecting HC Fab and Fc is indicated by the arrow. **d** The IgA1P-IgA1 complex is shown with the bound HC monomer (both Fab and Fc). **e** Expansion of the active site with the bound IgA1 hinge. IgA1P catalytic residues are shown according to the colors in Fig. 1e. The scissile bond is between P227 and T228 (residues shown in blue).

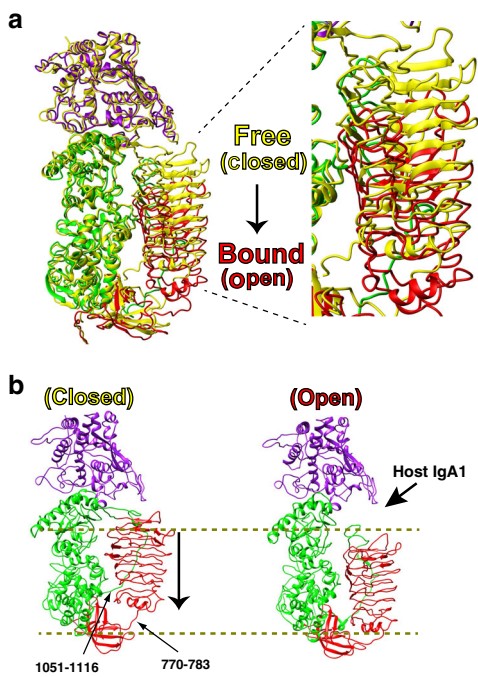

**Fig. 3 Structural rearrangement of IgA1P upon IgA1 binding.**
**a** Superposition of IgA1P in both free (yellow) and bound (colored by domain as in Fig. 1d) states. A least squares superposition of IgA1P residues 1115–1963 was used for all C-alpha atoms, illustrating the large 10 Å shift of the entire NTD β-helix. The unbound state is referred to as "closed" and the bound state as "open". **b** The 10 Å shift of the NTD β-helix is illustrated by both the free (left, closed state) and bound IgA1P (right, open state).

mechanism has been proposed for the evolutionarily distinct serine-type IgA1P from *Haemophilus influenzae* IgA1P[24], but our structure of the *S. pneumoniae* IgA1P/IgA1 complex here provides direct experimental proof that the metalloprotease subgroup of IgA1Ps must undergo such a conformational change to facilitate substrate binding.

**Blocking IgA1P active-site-gating through mAb binding.** Considering the clinical potential of utilizing immunogenic regions of IgA1P to develop vaccines preventing *S. pneumoniae* infection[6,7], we present here evidence validating that such a strategy has the potential to block IgA1P activity. A comparison of the neutralizing activity of a previously developed mAb used to detect secreted IgA1P[21], referred to as mAb #1, and a currently produced mAb, referred to as mAb #2, reveals that the latter potently blocks IgA1P cleavage of IgA1 and this occurs at stoichiometric concentrations (Fig. 4a). Moreover, this neutralizing mAb also blocks IgA1 substrate binding (Fig. 4b). A 3D reconstruction of IgA1P in the presence of the neutralizing mAb Fab fragment (mAb #2) resulted in a 4.8 Å resolution map (Resolution shells are shown in Fig. 4c and structural parameters and refinement statistics in Supplementary Fig. 2 and Supplementary Table 1), which facilitated an initial rigid body fit of the mAb #2 Fab with likely delineation of the individual VL-CL (variable light-constant light) and VH-CH1 (variable heavy-constant heavy 1) chains (Supplementary Fig. 5). The final refined model reveals that the neutralizing mAb simultaneously engages both the IgA1P NTD and MD domains (Fig. 4d, e), which confines IgA1P to its closed conformation through intimate contacts with both domains (Fig. 4f, g) and thereby occludes substrate binding. Thus, these studies provide a proof-of-concept that IgA1P activity may be blocked by mAbs.

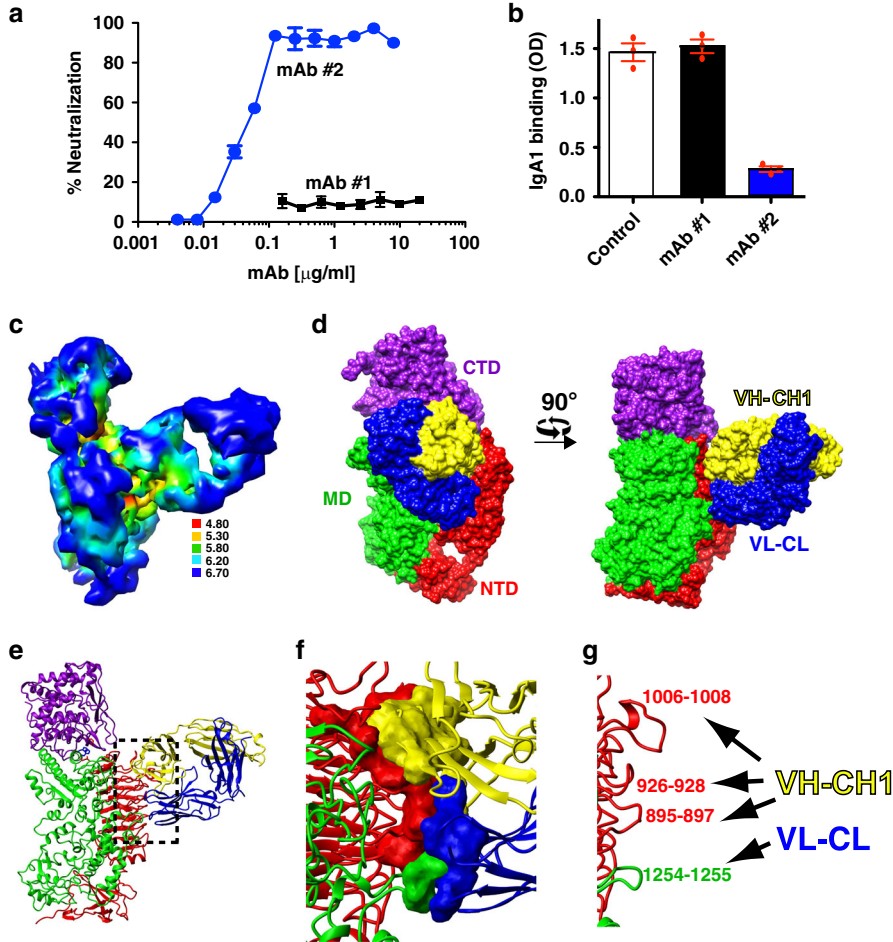

**Fig. 4 Cryo-EM structure of an *S. pneumoniae* IgA1P/mAb complex. a** ELISA-based neutralization assay comparing the blocking activity of two mAbs produced against IgA1P to cleave IgA1. The mAb #1 (black) previously developed does not neutralize IgA1P activity[21] and the currently characterized mAb #2 (blue) does neutralize IgA1P activity. IgA1P residues 154–1963 was incubated at 286 ng/ml (1.3 nM) with varying concentrations of the mAbs. For reference, 1 µg/ml of each mAb is approximately 6.7 nM estimating the molecular weight as 150 kDa. Data are an average of $n = 4$ biological replicates with the standard error of the mean also shown. **b** *S. pneumoniae* IgA1P-E1605A binds to IgA1 (white), which is not blocked by mAb #1 (black) but is blocked by mAb #2 (blue). IgA1P-E1605A was plated at 1 µg/ml and incubated with either buffer or 2 µg/ml mAb. Data are an average of $n = 3$ biological replicates shown as a dot blot along with the standard error of the mean also shown. **c** 3D reconstruction of the *S. pneumoniae* IgA1P in complex with IgA1 colored according to local resolution estimates (units in Å). **d** Surface representation of the IgA1P/mAb complex (PDB ID: 7JGJ) with IgA1P shown in a similar orientation as Fig. 1 (left) and rotated 90° (right). The complex comprises IgA1P (colored as in Fig. 1) and both the mAb VL-CL (blue) and mAb VH-CH1 (yellow). **e** Ribbon structure of the IgA1P/mAb complex with the box highlighting the interaction regions. **f** Expansion of the interaction site with residues that form a contact surface within 3 Å shown. **g** Specific interacting loops of IgA1P targeted by the mAb are shown. Source data for panels **a** and **b** are provided as a Source Data File.

## Discussion

We have determined the structure of a member of the M26 metalloprotease IgA1P subfamily of bacterial IgA1Ps both in isolation as well as in complex with its human IgA1 substrate and solved its structure with a neutralizing mAb that provides insight into blocking *S. pneumoniae* IgA1P activity. Other than the IgA1P NTD that comprises a β-helix identified in several bacterial proteins[23], both the IgA1P MD and CTD exhibit no structural similarity to any other previously solved proteins, much less any other metalloprotease. Despite this architecture, catalytic residues contributed by both the IgA1P MD and CTD that include H1604, E1605, H1608, and E1628 are directly comparable in their structural positioning to the same active site residues within the prototypical metalloprotease, thermolysin[25]. The most striking discovery here is how the modular nature of this unique IgA1P facilitates a conformational change that acts as an active-site gate. Specifically, this conformational change can be described as "closed" for the free enzyme and "open" to allow for

IgA1 substrate binding, whereby the mAb occludes substrate binding by binding the IgA1P in the "closed" conformation (Fig. 5). Our studies therefore provide the underlying mechanism by which this unique class of IgA1Ps engages IgA1 and reveals how IgA1Ps can broadly engage IgA1 through contacts with the constant region of the substrate IgA1 HC.

The complex structure with IgA1 also provides an understanding for how IgA1P can cleave the secreted forms of IgA1 (sIgA1) that are joined by the J-chain. Specifically, the Fc portions of each IgA1 that are coupled by the J-chain leave the IgA1 Fc binding site for IgA1P accessible for cleavage, as shown here by superimposing the bound IgA1 Fc within the IgA1P/IgA1 complex onto the recently solved structure of one of the sIgA1 Fc regions (Fig. 5)[29].

Considering the sequence similarity of the *S. pneumoniae* IgA1P to many other proteins found within invasive bacteria and across multiple strains (Supplementary Fig. 6a), these studies provide an important foundation for understanding how multiple

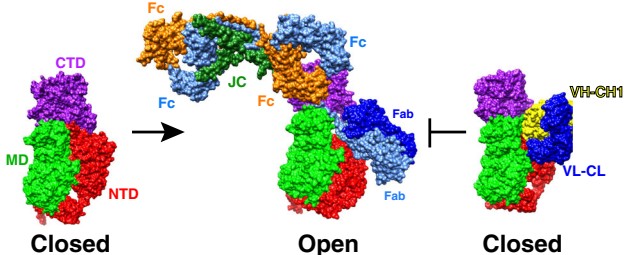

**Fig. 5 The structural basis of IgA1P cleavage and mAb neutralization.**
Models derived from all three cryo-EM reconstructions solved here are
shown. Left: IgA1P residues 665–1963 alone (PDB accession 6XJB). Middle:
The structure of IgA1P-E1605A in complex with human IgA1 is shown (PDB
accession 6XJA). The Fc portion of this structural model is superimposed
on one of the Fc regions within the recently solved cryo-EM structure of the
sIgA1 dimer[29]. The J-chain (JC) linking the two Fc portions of sIgA1 is also
shown (dark green). Right: IgA1P bound to the neutralizing mAb is shown in
a similar orientation to illustrate how this mAb occludes substrate binding.

opportunistic pathogens that employ similar IgA1 metallo-
proteases cleave host IgA1 (Supplementary Fig. 6b). While our cryo-
EM models demonstrate the specific site of the current neu-
tralizing mAb is not completely conserved across such strains,
there are several highly conserved epitopes that could serve for
the creation of broad-spectrum vaccines or multiple exposed
epitopes can now be developed to block multiple strains. Lastly,
the similarity in the predicted domain structure of this IgA1P in
comparison to proteases in the ZmpB and ZmpC classes of
secreted metalloproteases suggests that the structural data pro-
vided here can generally inform upon the mechanisms of cleavage
mediated by these related enzymes with possess divergent sub-
strate selectivity[30].

## Methods

**Protein expression and purification.** All *S. pneumoniae* IgA1P constructs of the
full catalytic region were engineered and purified similar to that previously
described for the mature IgA1P (residues 154–1963), with the exception that the
MBP and 6xHis tags were swapped from their original positions[21]. The kanamycin-
resistant plasmid, pJ401 (DNA2.0/Atum), was used as the backbone with the open
reading frame coding for an N-terminal MBP, a thrombin cleavage site, IgA1P
residues 665–1963 (or variants thereof described in the main text), and a C-
terminal 6xHis tag. A typical growth comprised 4 L of Luria Broth (LB). Protein
was expressed in *E. coli* BL21(DE3) and induced at an OD$_{600}$ of 0.6–0.9 with 1 mM
IPTG at 37 °C for 3 h before harvesting. Cells were lysed via sonication and lysed
supernatants were applied to amylose resin using amylose buffer (20 mM Tris, pH
7.4, and 200 mM NaCl) on an AKTA prime FPLC system (GE Healthcare). The
protein was eluted using 20% glucose in amylose buffer, applied to Ni-affinity resin
(Sigma) using Ni buffer (50 mM phosphate buffer, pH 7.5, 500 mM NaCl, and
10 mM imidazole), and eluted using Ni buffer supplemented with 0.4 M imidazole.
The eluent was then concentrated, cleaved with thrombin (Sigma) at room tem-
perature overnight, and diluted for reapplication to amylose resin to strip
remaining MBP or uncleaved fusion. Finally, amylose-resin flow through was
concentrated and applied onto an S200 size exclusion column (120 ml, GE
Healthcare) equilibrated in 20 mM HEPES, pH 7.4, and 150 mM NaCl.

The IgA1P G5 domain and CTD were expressed and purified as previously
described[22]. The IgA1P NTD was engineered into the pJ401 plasmid and expressed
and purified using the same method previously described[21,22]: an N-terminal 6xHis
tag was followed by a thrombin cleavage site and IgA1P residues 665–1010. All
individual domains were purified over Ni-affinity resin, concentrated for thrombin
cleavage of their N-terminal 6xHis tag, cleaved with thrombin at room temperature
overnight, and applied to a S75 size exclusion column (120 ml, GE Healthcare).

IgA1 kappa was purchased commercially (myBioscience; San Diego, CA) and
further purified over a preparative S200 size exclusion column in 20 mM HEPES,
pH 7.4, and 150 mM NaCl in order to specifically select for monomeric IgA1.

The murine monoclonal antibodies used here (mAb #2) and previously
described (mAb #1)[21] were produced as follows. Balb C female mice were
immunized intraperitoneally with 100 µg of purified full-length protease with
Freund's adjuvant twice and boosted with protein alone at 3 weeks. Protease-
specific IgG in sera collected 7 days after the third injection was detected by ELISA
as described below. Spleen cells were fused with PEG and SP2/0 myeloma cells.
Hybridomas producing protease-specific IgG were subcloned twice and IgG
purified on Protein G columns (Pierce-ThermoFisher; Waltham, MA). Clones

reactive with the protease by ELISA were screened and selected based on optimal
neutralization as described below (neutralizing activity per µg of protease-specific
IgG). The Fab fragment for mAb #2 used here for cryo-EM in complex with IgA1P
residues 665–1963 was excised from the full antibody by use of immobilized Ficin
(Thermo Scientific) and diluted using PBS. The "Standardized Protocol for
Production of Monoclonal Antibodies in Mice", protocol # 104513(12)1C, was
approved by the Animal Care and Use Committee, accredited by the Association
for Assessment and Accreditation of Laboratory Animal Care—File Number
00235. We have complied with all relevant ethical regulations for animal testing
and research.

**Cryo-EM sample preparation and data collection.** For *S. pneumoniae* IgA1P
alone, purified *S. pneumoniae* IgA1P (residues 665–1963) was concentrated to
~3.4 µM (0.5 mg/ml) in 20 mM HEPES, pH 7.4, and 150 mM NaCl. For the
*S. pneumoniae* IgA1P/IgA1 complex, purified *S. pneumoniae* IgA1P-E1605A
(residues 665–1963) was concentrated to 90 µM and was incubated with 30 µM of
the IgA1 substrate in the presence of 1 mM EDTA. This IgA1P/substrate complex
was further purified in 20 mM HEPES, pH 7.4, and 150 mM NaCl, 1 mM EDTA
over an analytical Superdex 200 (GE Healthcare) in order to enrich the 1:1 complex
(Supplementary Fig. 1d). The same procedure was used for the *S. pneumoniae*
IgA1P/mAb (the Fab fragment) complex but using wild type IgA1P instead of the
IgA1P-E1605A mutant and in the absence of EDTA. For grid preparations, 3 µl of
0.1–0.2 mg/ml sample was applied to plasma-cleaned C-flat holy carbon grids
(1.2/1.3, 400 mesh) and frozen using a Vitrobot Mark IV (Thermo Fisher Scien-
tific), with the environmental chamber set at 100% humidity and 4 °C. The grids
were blotted for 2.5–3.0 s and then flash frozen in liquid-nitrogen-cooled liquid
ethane. A full description of the cryo-EM data collection can be found in Sup-
plementary Table 1.

**Cryo-EM data processing and structural modeling.** Data for the free *S. pneu-
moniae* IgA1P (residues 665–1963) were processed in RELION[31] and that of the *S.
pneumoniae* IgA1P-E1605A/IgA1 complex were processed in both RELION and
cryoSPARC[23,31]. Models for both complexes were built using Coot[32]. For IgA1P
alone, the movies were motion corrected using MotionCor2[33] and their contrast
transfer functions were estimated using Gctf[34]. Micrographs with fit resolution of
worse than 6 Å were discarded. ~1500 particles were manually picked and classified
into 10 classes. Meaningful 5 representative class averages were selected and used
for template picking. The particles were extracted with a box size of 336 × 336
pixels and subjected to 2D classification. Classes with clear visible secondary fea-
tures were selected and subjected to ab initio reconstruction, 3D classification, and
refinement. After particle polishing and CTF refinement, the final model for
unbound IgA1P (residues 665–1963) was estimated as 3.77 Å resolution based on
the gold-standard Fourier shell correlation (FSC) measurement. The sequence of
IgA1P residues 665–1963 was analyzed by multiple secondary structure prediction
servers as well as the I-TASSER 3D structure prediction server[35]. This analysis
revealed the NTD comprised the β-helix domain (795–1050), which served as the
starting point of our model building. For the rest of the IgA1P model, the main-
chain was traced with the help of MAINMAST[36]. The final model was complete in
Coot[32], followed by the real-space refinement in PHENIX[37].

For the IgA1P-E1605A/IgA1 complex, a similar process protocol was carried
out, except that the particles were binned twice to make the final pixel size 1.06 Å/
px. The resolution of the final reconstruction for the IgA1P-E1605A/IgA1 complex
was estimated as 3.5 Å in RELION with similar resolution in cryoSPARC. This map
facilitated rigid body fitting using the free IgA1P residues 665–1963 solved above,
the substrate IgA1 Fc (PDB accession number 1OW0), and the substrate IgA1 Fab
(PDB accession 3M8O), which was further refined (Fig. 2). As the IgA1 used was
polyclonal, all variable residues within the Fab region of both its HC and LC were
modeled as alanine residues (Supplementary Fig. 3). This did not alter the model of
IgA1P as IgA1P only makes contacts with the IgA1 constant region.

For the IgA1P/mAb complex, processing was identical to that of the free
protease. The solution of the final reconstruction for the IgA1P/mAb complex was
4.8 Å. The free IgA1P residues 674–1963 and a model of the mAb constructed in
SWISS-MODEL was placed within this 3D reconstruction using Chimera. For this
initial mAb model, the mAb HC and LC were sequenced and PDB accession
3RHW and 4HXA were found to be the most similar, respectively. Models for these
two chains were produced separately in SWISS-MODEL, superimposed upon that
of the full 3RHW Fab, and then placed within the 3D reconstruction within
Chimera.

All data were collected with Leginon 3.3. The local resolution estimations of
both reconstructions were performed using RELION's own postprocessing
protocol and models were refined using both Phenix[37] and Coot[32].

**NMR sample preparation and data collection.** All NMR data were collected on a
Varian 900 equipped with a cryoprobe. Samples were produced as previously
described using $^{15}$N M9 minimal media for the *S. pneumoniae* IgA1P G5 domain
(residues 312–393)[22] and $^{15}$N,$^{2}$H M9 minimal media for both the CTD (residues
1611–1963)[21] and the NTD (residues 665–1010). Final size exclusion was per-
formed in NMR buffer (50 mM Na$_2$HPO$_4$, pH 6.5, and 150 mM NaCl).

**Protease neutralization assay**. 96 well microtiter plates were coated with anti-IgA1 CH3 (catalog number ab17747, Abcam; Cambridge, MA) (2 mg/ml) overnight at 4 °C, washed, and blocked with PBST-BSA for 2 h at room temperature. Wild type IgA1P residues 154–1963 (286 ng/ml; 1.3 nM) was incubated overnight with the indicated doses of the neutralizing mAb at 4 °C. Human IgA1 substrate (catalog number MBS318189, myBioscience; San Diego, CA) alone or with equal volumes of the mAb/IgA1P mixture were incubated for 1 hr at 37 °C and added to the plates for 2 h at room temperature. After washing, alkaline phosphatase-labeled goat anti-human Kappa (catalog number A38913, Sigma; St. Louis, MO) or goat HRP-labeled anti-human IgA CH3 (Abcam; Cambridge, MA) was added for an hour at room temperature. Intact or cleaved IgA1 was detected by the presence or absence of bound kappa light chain using alkaline phosphatase-labeled goat anti-human Kappa (Sigma; St. Louis, MO), which is lost by IgA1 cleavage. IgA1 binding was normalized by HRP-labeled goat anti-human Fcα. Both are read at 405 nm in separate wells. Plates were either developed with p-nitrophenyl phosphate substrate-AP (Sigma) or ABTS-HRP, respectively.

**IgA1 binding assay**. 96 well Nunc Maxisorp plates were coated with 1 μg/ml (4.6 nM) IgA1P-E1605A residues 154–1963 in PBS overnight at 4 °C, blocked for 2 h with PBS-Tween20-Thimersol-0.5% BSA(PBS-TT-BSA) and incubated with 100 ng/ml human IgA1 (myBioscience; San Diego, CA), which was followed by the addition of 2 μg/ml mAb (14.6 nM). Human IgA1 that bound to the protease was detected with HRP-labeled goat anti-human IgA, developed with peroxide and ABTS, and optical densities measured at 405 nm with Veramax ELISA plate reader.

**Reporting summary**. Further information on research design is available in the Nature Research Reporting Summary linked to this article.

## Data availability

The cryo-EM maps are deposited in the Electron Microscopy Data Bank under accession codes "EMD-22205" (IgA1P residues 665–1963), "EMD-22204" (IgA1P residues 665–1963 with the single E1605A mutation in complex with IgA1), "EMD-22328" (IgA1P residues 665–1963 in complex with the mAb). Structure coordinates are deposited at the Protein Data Bank with accession codes "6XJB" (IgA1P residues 665–1963), "6XJA" (IgA1P residues 665–1963 with the single E1605A mutation in complex with IgA1), "7JGJ" (IgA1P residues 665–1963 in complex with the mAb). Source data is provided with this paper.

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

## Acknowledgements

We thank the CU Cryo-EM Structural Biology Shared Resource Facility for data collection for IgA1P alone. We would like to thank Lori Sherman and the University of Colorado Cancer Center Protein Production/MoAB/Tissue Culture Shared Resource for their help

with monoclonal antibody production. Data collection of the IgA1P-E1605A/IgA1 complex was supported by NIH grant U24GM129547 and performed at the PNCC at OHSU and accessed through EMSL (grid.436923.9), a DOE Office of Science User Facility sponsored by the Office of Biological and Environmental Research. NMR spectra were collected at the University of Colorado Spectroscopy Facility supported by the University of Colorado Cancer Center NIH P30 CA046934. J.E. was supported by Veterans Affairs Research Service I01BX004320 and NIH R21 AI092468. H.Z. was supported by NIH R01 GM126626. E.Z.E. was supported by NIH R21 AI146295 and NSF application number 1807326.

## Author contributions

Z.W. and H.Z. determined the 3D reconstructions and Z.W. refined IgA1P alone. N.T., T.H. and E.Z.E. refined both complexes. Y.C.C. began the initial cloning of the IgA1P with E.Z.E., J.R. and E.J. produced the mAb with J.R. and E.Z.E. purifying the product. J.S.R. and E.Z.E. purified all proteins and complexes.

## Competing interests

The authors declare no competing interests.
