## [Peer Review File · Nature Communications]

REVIEWER COMMENTS

Reviewer #1 (Remarks to the Author):

IgA1P is a *Streptococcus pneumoniae* effector protease that cleaves host IgA1 to suppress host defense. Despite the fact that IgA1P has been discovered for a while, its structure and mechanism remains poorly understood due to its large size and low homology to other proteins. In this manuscript, the authors performed comprehensive structural analyses of IgA1P using cryo-EM, and determined several structures including the IgA1P/IgA1 complex. The work is unique and interesting, and most of the conclusions are sound (except for the points below). The main drawback of this study, in the current form, is the writing, which is very unsatisfactory, and somewhat diminishes the strength of this paper.

1. The structure of IgA1P is built ab initio using a 3.8 angstrom cryo-EM map, which is quite remarkable. The authors should include more information regarding the modeling process and the final structural model. Did the authors start from predicted domain structures generated by programs such as SWISS-MODEL or Rosetta, or really build everything from scratch? Regions that are disordered or built as a poly alanines perhaps should be mentioned. Supplementary Table 1 needs to be consolidated to include important information such as map-model CC. More importantly, a Zn ion is not present in the final coordinate—so Figure 1e is generated purely based on structural comparison with thermolysin. This should be specified in the figure legend.

2. The modeling of IgA1P/IgA1 complex is reasonable. One small thing: IgA1P-E1605A is used to prepare the complex, but the corresponding residue is still modeled as a Glu in the final coordinate. This should be corrected.

3. The analysis of IgA1P conformational change upon IgA1 binding is valid. However, the “linearly extrapolated structure” in the middle of Figure 3b is confusing and misleading. Molecular dynamics simulation needs to be performed to assess the transition state. The left and right panels are sufficient to simply illustrate the structural change.

4. The modeling of the IgA1P/mAb is somewhat troublesome. The map does indicate where the mAb roughly binds, however, I am not so sure that the authors can distinguish the heavy chain and light chain due to the low resolution. Ideally, the authors should collect more data and improve the resolution. If this cannot be easily achieved, the authors should tune down their analyses and do not specifically discuss HC and LC in places such as Figure 4 and Supplementary Figure 2f. In particular,

Supplementary Figure 2f appears to suggest that the interaction between the mAb and IgA1P is mainly mediated by the light chain, which is unusual for the antibody-antigen interactions.

5. Supplementary Figure 5 is an important piece of data, and should be sufficiently described and perhaps even presented in the main figure. In fact, the whole mAb section is inadequately presented. How is the mAb identified? Have the authors measured its binding affinity for IgA1P? Have the authors tested the neutralization activity of the mAb for the live *S. pneumoniae* bacteria, instead of pure IgA1P protein?

6. The Discussion section needs to be expanded. For example, how does IgA1P compare to other IgA proteases, or proteases in general? Only thermolysin is mentioned in the paper. How does IgA1P specifically target IgA, instead of IgG? Can IgA1P cleave IgA2? Also, *S. pneumoniae* dwells on the mucosal surface, so the natural target of IgA1P is likely mucosal IgA, which contains a IgA dimer in complex with J-chain and secretory piece (SIgA). Can the authors compare the IgA1P/ IgA1 structure with the recently determined core SIgA structure (Science, 2020; Cell Research, 2020), and comment whether the presence of J-chain and secretory piece has an impact on IgA1P binding? These topics are all relevant to this study, and should be discussed.

7. Supplementary Figure 2 is totally unacceptable. The texts are too small in panels a-c. Panel d: the model has hydrogen atoms—why? Panel e looks awful. This figure should contain information such as typical 2D classes, flow chart of EM image processing, 3D classes, angle distribution of particles...Densities should be shown for important regions (e.g., the interface) and representative areas for each structure. These are all important for the readers to evaluate the data and model quality.

Minor points:

1. FAB should be defined at the beginning. Also, Fab is more commonly used in the literature, instead of FAB.
2. Page 3, Line 83, “ful” is a typo for “full”? But “full” won’t make sense here either.
3. Figure 4 Legend, should be “IgA1P in complex with mAb” in a).
4. Also in Figure 4, FC is a typo for HC?

Reviewer #2 (Remarks to the Author):

In general this is an interesting study with some good structural insights into the interaction of IgA and IgA1P and how it may be blocked for therapeutic effect.

Figure 4 – it could be made clearer how the mAb stabilised structure relates to the active site gating mechanism and associated structure detailed in figure 3. (esp fig 3b).

It should be described that the mAb binds in the same region as the IgA. Therefore the likely mechanism is through direct orthosteric inhibition rather than any subtle orthosteric mode. I find the clamping analogy a little misleading in that regard. The mode of action of inhibition by the mAb should be made clearer. An overlay model of IgA and mAb on IgA1P might help illustrate this.

Supplementary line 31-33 suggests that the murine mAb was based from a reference (1). On review of Chi et al (2017) it is not clear what the antigen for immunisation was. In the context of the work in the current paper, it would be useful to understand the exact form of antigen that was used for this previous work given it has functional activity and recognises an interesting conformational inactive state.

Data in supplementary figure 5 looks convincing and provides evidence that the mAb blocks activity of IgA1P and that activity is through direct inhibition of substrate binding.

Line 179 in the main manuscript makes no reference to the fact that the 3D reconstruction is based on Cryo EM data.

Reviewer #3 (Remarks to the Author):

Wang et al present in their paper the structure of the prototypical IgA1 metalloprotease secreted by the pneumococcus and provide a model how this protease binds IgA1 prior to proteolysis. Active site residue substitution as well as chelating the catalytic Zn-ion allowed them to investigate co-

structures between the enzyme and the intact substrate by cryo-electron microscopy. Comparing the structures of free IgA1P and IgA1P bound to its substrate IgA1 suggests an active site gating mechanism including a large shift of the N-terminal domain. Moreover, the proteolytic activity of IgA1P was inhibited with a mAb that prevents this conformational change needed for substrate binding. The authors claim that this inhibition would “block infection”. To support this claim, the infection model used by Janoff et al (Mucosal Immunology, 2013) where mice were passively immunized with pneumococcus specific human IgA1 prior to infection could be used. The statement that neutralization of this single immune evasion factor blocks infection is probably exaggerated and needs therefor experimental prove or has to be toned down.

Minor points:

As the authors suggest that IgA1 metalloproteases provide a platform for broad-spectrum vaccines, they should show that IgA1P is conserved among different endemic pneumococcal strains of different serotypes and also among other pathogens encoding IgA1 metalloproteases (e.g. *S. sanguinis* and *S. suis*). Especially conservation of the antigenic regions where the mAb binds (highlighted in Fig. 4e) should be investigated to support this claim. Targeting bacterial antibody degrading enzymes by vaccines has indeed shown promising results in several vaccine studies against pathogens of veterinary importance.

It would be interesting to know if cleavage of dimeric substrate IgA1 (consisting of two heavy chains) is a one step or a two-step process. Monitoring cleavage during early time points by non-reducing SDS-PAGE (separating intact IgA, sIgA, Fab and Fc fragments) could provide further insights into the cleavage mechanism.

Uncommon nomenclature for different parts of immunoglobulins are used e.g. FAB instead of Fab, HC-Fc instead of Fc (as Fc only consists of HC), LC-FAB instead of LC (as LC only is part of Fab). Referring to the domains such as CH1, VL etc might also help the reader (e.g in Fig. S3). Please use standard nomenclature throughout the entire manuscript.

Line 54-59: References 4-10 describing the epidemiology of pneumococcal infections and burden to global health as well as need for vaccines are not up to date. The authors should refer to more current studies.

Line 64: Referring to the M26 family in the Merops peptidases database might help the reader.

Line 76: Reference 3 is not presenting experimental data supporting the foregoing statement.

Line 83: full not ful

Line 84 or supplementary information line 4: The authors should here or in the material and methods refer to an UniProt accession number or any other identifier of the studied IgA1P as different pneumococcal strains have slightly different aa sequences. Also any information about the pneumococcal strain that this sequence is originating from is missing. This information can neither be found in reference 1 of the supplementary data.

Line 88: Activities cannot be compared at time points where the substrate is already depleted. Shorter time points or lower enzyme concentrations have to be used for semi-quantitative comparisons. To compare activities, I would suggest to find conditions where <50% of the initial substrate is degraded.

Line 177 and Supplementary data, line 31: Was this monoclonal Ab identified during the current study in the same way as in reference 1 of the supplementary information or was this mAb already identified in the previous study? More details should be given if this is a novel mAb. If it is the same mAb as published by Chi et al., this should be more clearly stated.

Line 267-269: Ying-Chih Chi is not included in the author contributions.

Fig. 2d: It is not clear what the dashed box is referring to. It does not correspond to the expansion shown in Fig. 2e.

Fig. 4b, e: HC-mAb is mistakenly labelled as Fc-mAb contradicting the figure text. Labelling the blue as VH-CH1 and the yellow as VL-CL would be preferable.

Supplementary data, line 98: How was neutralization percentage calculated from these two measurements? This is not clear from the text.

Supplementary data, line 98: IgA1P-E1605A instead of E1605

Fig. S1d: Y-axes labelling as "intensity" is unclear. Is it absorption at 280 nm?

Fig. S5a: Concentration of IgA1P should be mentioned and what the molar ratios between IgA1P and the mAb are. Otherwise the potency of the mAb cannot be judged. It is moreover unclear what the error bars show.

Fig. S5b: Scale is missing.

We thank the reviewers for their thoughtful assessments and kind words regarding these studies. Changes to both the main manuscript and supplementary data are in red and we have included three new figures to further address their comments as well (Figure 5, Supplementary Figure 5, Supplementary Figure 6).

Reviewer #1 (Remarks to the Author):

1. The structure of IgA1P is built ab initio using a 3.8 angstrom cryo-EM map, which is quite remarkable. The authors should include more information regarding the modeling process and the final structural model. Did the authors start from predicted domain structures generated by programs such as SWISS-MODEL or Rosetta, or really build everything from scratch? Regions that are disordered or built as a poly alanines perhaps should be mentioned. Supplementary Table 1 needs to be consolidated to include important information such as map-model CC. More importantly, a Zn ion is not present in the final coordinate—so Figure 1e is generated purely based on structural comparison with thermolysin. This should be specified in the figure legend.

Response: We apologize for the lack of details for model building. We have now elaborated on the specifics of model building the IgA1P within Methods whereby we used an initial model of the beta-helix to initiate the remaining ab initio model. The specific alanine positions are shown in Supplementary Figure 3a (HC) and Figure 3b (LC). We apologize for any confusion, as these were not used to fit disordered regions but rather variable residues and this is stated within the main text. We have edited the data to also include map-model CC. We have now also stated in the figure (Figure 1) that the Zn was positioned in accordance with the super-positioning of the thermolysin active site for the four conserved active site residues.

2. The modeling of IgA1P/IgA1 complex is reasonable. One small thing: IgA1P-E1605A is used to prepare the complex, but the corresponding residue is still modeled as a Glu in the final coordinate. This should be corrected.

Response: We have replaced the submission of 6XJA with the appropriate E1605A mutation and have noted that the catalytic E was placed at this position for Figure 1 within the legend.

3. The analysis of IgA1P conformational change upon IgA1 binding is valid. However, the “linearly extrapolated structure” in the middle of Figure 3b is confusing and misleading. Molecular dynamics simulation needs to be performed to assess the transition state. The left and right panels are sufficient to simply illustrate the structural change.

Response: We agree that this extrapolated structure did not add to the figure and have therefore removed it in accordance with the reviewer’s suggestion.

4. The modeling of the IgA1P/mAb is somewhat troublesome. The map does indicate where the mAb roughly binds, however, I am not so sure that the authors can distinguish the heavy chain and light chain due to the low resolution. Ideally, the authors should collect more data and improve the resolution. If this cannot be easily achieved, the authors should tune down their analyses and do not specifically discuss HC and LC in places such as Figure 4 and Supplementary Figure 2f. In particular, Supplementary Figure 2f appears to suggest that the interaction between the mAb and IgA1P is mainly mediated by the light chain, which is unusual for the antibody-antigen interactions.

Response: This is indeed an important point to convey why we modeled this way. We could in fact discern the likely mode of binding regarding the neutralizing mAb VH-CH1 and VL-CL due to the fact that these domains are not completely symmetrical. In order to illustrate this likely fit, we have now provided an additional supplementary figure, Supplementary Figure. 5, which depicts the two modes of fitting VH-CH1 and VL-CL.

5. Supplementary Figure 5 is an important piece of data, and should be sufficiently described and perhaps even presented in the main figure. In fact, the whole mAb section is inadequately presented. How is the mAb identified? Have the authors measured its binding affinity for IgA1P? Have the authors tested the neutralization activity of the mAb for the live *S. pneumoniae* bacteria, instead of pure IgA1P protein?

Response: We apologize for this lack of clarity and we have now provided a comparative analysis between the previously published mAb (Chi, 2017), referred to here as mAb #1, and the new mAb presented here, mAb #2. This analysis illustrates why we identified the current mAb as ideal for these structural studies, as this mAb was neutralizing (now Figure 4a) and blocked IgA1 binding (now Figure 4b). Namely, mAb #1 did not neutralize IgA1P activity despite successfully identifying secreted IgA1P in our previous study (Chi et al, 2016), while mAb #2 does neutralize activity in a potent manner (now, Figure 4.a). We have also provided the molar concentrations within the figure legend. Binding affinities have not been specifically determined, as our primary focus was on identifying a neutralizing mAb that could serve as a model for how to block IgA1P activity. However, the mAb #2 blocks activity at stoichiometric concentrations at nanomolar concentrations and the complex co-migrates on size-exclusion chromatography and thus, binding is likely within this range or tighter. Regarding live *S. pneumoniae* bacteria, murine studies within the Janoff lab will be under development once funding can be secured.

6. The Discussion section needs to be expanded. For example, how does IgA1P compare to other IgA proteases, or proteases in general? Only thermolysin is mentioned in the paper. How does IgA1P specifically target IgA, instead of IgG? Can IgA1P cleave IgA2? Also, *S. pneumoniae* dwells on the mucosal surface, so the natural target of IgA1P is likely mucosal IgA, which contains a IgA dimer in complex with J-chain and secretory piece (sIgA). Can the authors compare the IgA1P/ IgA1 structure with the recently determined core sIgA structure (Science, 2020; Cell Research, 2020), and comment whether the presence of J-chain and secretory piece has an impact on IgA1P binding? These topics are all relevant to this study, and should be discussed.

Response: We have expanded the Discussion section to more completely evaluate our findings and thank the reviewer for this suggestion. We have included a brief discussion regarding the lack of structural similarity between the IgA1P MD and CTD that forms this unique metalloprotease active site and further described the finding that, despite this unique structure, the same four catalytic residues as thermolysin can be structurally superimposed. IgA1P is specific for IgA1 due to its extended linker relative to IgA2 and IgGs and we have now included a reference from Woof and Russell (2011) that reviews the body of literature describing such specificity within the first mention of IgA1P function. We agree with the reviewer that it is also of fundamental importance to address how IgA1P can readily cleave sIgA. Thus, in expanding the Discussion, we have now included an additional figure (now Figure 5) to specifically summarize our results and address how the IgA1P can cleave sIgA. This figure also provides a structural comparison with similar orientations to illustrate how the neutralizing mAb occludes IgA1.

7. Supplementary Figure 2 is totally unacceptable. The texts are too small in panels a-c. Panel d: the model has hydrogen atoms—why? Panel e looks awful. This figure should contain

information such as typical 2D classes, flow chart of EM image processing, 3D classes, angle distribution of particles...Densities should be shown for important regions (e.g., the interface) and representative areas for each structure. These are all important for the readers to evaluate the data and model quality.

Response: We apologize for the small print for the FSCs as exported from RELION and cryoSPARC and have corrected this. We have provided further details explicitly describing the model building, especially in regard to the IgA1P itself that began from initial models of the beta helix within the new Methods. We have remade Supplementary Figure 2 to also provide representative 2D class averages and new figures to illustrate the density of each model. For the old panel e (that is now panel h), we had solely included the IgA1 HC linker to illustrate its density, as the density is quite crowded within this region. However, we have replaced this with the entire region and apologize for any lack of clarity for the former view.

Reviewer #2 (Remarks to the Author):

Figure 4 – it could be made clearer how the mAb stabilised structure relates to the active site gating mechanism and associated structure detailed in figure 3. (esp fig 3b).

It should be described that the mAb binds in the same region as the IgA. Therefore the likely mechanism is through direct orthosteric inhibition rather than any subtle orthosteric mode. I find the clamping analogy a little misleading in that regard. The mode of action of inhibition by the mAb should be made clearer. An overlay model of IgA and mAb on IgA1P might help illustrate this.

Response: We did not mean to imply that the mAb solely functions through a clamping mechanism that is independent of steric occlusion of the IgA1 substrate (meaning, allosteric). The reviewer is completely correct in that the mAb interaction to both the IgA1P NTD and MD domains also occludes substrate binding. We have thus clarified this and further show via a structural comparison in our Discussion section (new Figure 5) that the reviewer is correct. We have also removed the “clamp” analogy as well and replaced this description regarding the gating mechanism as “closed” versus “open”. We believe that this may clarify how the mAb keeps the IgA1P closed and simultaneously occludes substrate binding and apologize for any lack of clarity.

Supplementary line 31-33 suggests that the murine mAb was based from a reference (1). On review of Chi et al (2017) it is not clear what the antigen for immunisation was. In the context of the work in the current paper, it would be useful to understand the exact form of antigen that was used for this previous work given it has functional activity and recognises an interesting conformational inactive state.

Response: We apologize for this confusion and we have now explicitly reported that this neutralizing mAb is new and that the former mAb used in Chi et al. (2017) to identify secreted forms of the IgA1P did not neutralize IgA1P activity. We refer to the first mAb that does not neutralize IgA1P activity as mAb #1 and the new mAb that does neutralize IgA1P activity as mAb #2 and we now provide the data to illustrate the comparative analysis between the two mAbs within the new Figure 4a,b.

Data in supplementary figure 5 looks convincing and provides evidence that the mAb blocks activity of IgA1P and that activity is through direct inhibition of substrate binding.

Response: We have put this figure that evaluates the neutralizing activity and occlusion of IgA1 binding into Figure 4 in compliance with another reviewer’s request and we have added the comparative data for our previously utilized mAb, as described above.

Line 179 in the main manuscript makes no reference to the fact that the 3D reconstruction is based on Cryo EM data.

Response: This has been corrected and we apologize for any lack of clarity here.

Reviewer #3 (Remarks to the Author):

The statement that neutralization of this single immune evasion factor blocks infection is probably exaggerated and needs therefor experimental prove or has to be toned down.

Response: While the IgA1 protease is considered a virulence factor, we completely understand the reviewer's concerns here and have instead described the implications of this work as molecular studies that can be used to "potentially block infection". We have also removed the language that stated that "we could block infection at the host interface". Our hope is that these studies will be used to generate a greater understanding of the pathogen/host interface and have now identified exposed regions that can be used as vaccine candidates going forward. We apologize for over-stating the importance.

Minor points:

As the authors suggest that IgA1 metalloproteases provide a platform for broad-spectrum vaccines, they should show that IgA1P is conserved among different endemic pneumococcal strains of different serotypes and also among other pathogens encoding IgA1 metalloproteases. Especially conservation of the antigenic regions where the mAb binds (highlighted in Fig. 4e) should be investigated to support this claim. Targeting bacterial antibody degrading enzymes by vaccines has indeed shown promising results in several vaccine studies against pathogens of veterinary importance.

Response: This is a very good point and we have therefore provided an additional supplementary figure that highlights the sequence similarity of *S. pneumoniae* IgA1P with other bacterial proteins (Supplementary Figure 6). For example, Supplementary Figure 6a includes a general sequence comparison that uses all available sequences to provide a quantitative value ascribed to each position within IgA1P that is reported here. In Supplementary Figure 6b we specifically map the sequence similarity to the other two strains for which we specifically compare this IgA1P to, which includes both *Streptococcus sanguinis* and to *Streptococcus oralis* IgA1Ps. We have not mapped the *Streptococcus suis* sequence similarity onto our structure within Figure 6b, as there is evidence to indicate that this protease is not an IgA1 protease but likely has other host targets similar to ZmpB (Dumesnil et al, Vet. Res., 2018).

It would be interesting to know if cleavage of dimeric substrate IgA1 (consisting of two heavy chains) is a one step or a two-step process. Monitoring cleavage during early time points by non-reducing SDS-PAGE (separating intact IgA, sIgA, Fab and Fc fragments) could provide further insights into the cleavage mechanism.

Response: The reviewer brings up an interesting point in that IgA1 exists as both a monomeric species but also as multimeric forms linked by the J-chain. For our cryo-EM studies, we purified the monomeric IgA1 from a commercial source in order to then concentrate this species for co-migration/purification with the enzyme (IgA1P-E1605A (Supplementary Figure 1d), thereby allowing us to control the specific complex and avoid inhomogeneous mixtures that would comprise these other forms. We therefore provide an explicit superposition of our IgA1P/IgA1 with that of the recently determined secreted dimeric form (new Figure 5) within the Discussion in order to provide the reader with the underlying reason why these forms are all amenable to IgA1P cleavage. Namely, the IgA1 Fc binding sites are still accessible to IgA1P and the remainder of the IgA1 substrate is as well.

Uncommon nomenclature for different parts of immunoglobulins are used e.g. FAB instead of Fab, HC-Fc instead of Fc (as Fc only consists of HC), LC-FAB instead of LC (as LC only is part

of Fab). Referring to the domains such as CH1, VL etc might also help the reader (e.g in Fig. S3). Please use standard nomenclature throughout the entire manuscript.

Response: We apologize, as our original intention was to clarify the fact that only the HC is cleaved by IgA1 and is shown in Figure 2d. We have replaced these with Fab and Fc to describe the substrate now, as the reviewer is correct in that this should be clear.

Line 54-59: References 4-10 describing the epidemiology of pneumococcal infections and burden to global health as well as need for vaccines are not up to date. The authors should refer to more current studies.

Response: We have replaced these references.

Line 64: Referring to the M26 family in the Merops peptidases database might help the reader.

Response: A very good point considering that this IgA1P is the prototypical member of the entire M26 family and this has been added accordingly.

Line 76: Reference 3 is not presenting experimental data supporting the foregoing statement.

Response: This reference has been replaced with two new references that describe explicit in vivo studies where this removal was first described as well as a general review for the fact that the IgA1 Fc receptor is recognized by its receptor on neutrophils (CD89).

Line 83: full not ful

Response: We have corrected this and apologize.

Line 84 or supplementary information line 4: The authors should here or in the material and methods refer to an UniProt accession number or any other identifier of the studied IgA1P as different pneumococcal strains have slightly different aa sequences. Also any information about the pneumococcal strain that this sequence is originating from is missing. This information can neither be found in reference 1 of the supplementary data.

Response: The reviewer brings up a very good point regarding the variation in strains and we apologize for not including this information. We have now included both the UniProt accession number and the NCBI accession numbers that corresponds to the IgA1 protease studied here that is found within two common strains, D39 and R6 strains, when first discussing the purification. We have also provided the UniProt accession numbers when comparing other strains in the new Supplementary Figure 6. We understand that this is critical when describing the specifics of each protein and we thank the reviewer for pointing this out.

Line 88: Activities cannot be compared at time points were the substrate is already depleted. Shorter time points or lower enzyme concentrations have to be used for semi-quantitative comparisons. To compare activities, I would suggest to find conditions were <50% of the initial substrate is degraded.

Response: We have omitted the word “activities”, as the reviewer is correct in that we are simply monitoring cleavage of IgA1 that is assessed by the qualitative comparisons through SDS-PAGE gels. Quantitative catalytic assays have been unsuccessfully sought for this IgA1P for quite some time, which is likely due to the very findings that we have presented here. Namely,

active site gating likely requires the fully intact substrate and thus, smaller peptide substrates have historically not been amenable to cleavage in order to develop activity assays that monitor standard Michaelis-Menten kinetics.

Line 177 and Supplementary data, line 31: Was this monoclonal Ab identified during the current study in the same way as in reference 1 of the supplementary information or was this mAb already identified in the previous study? More details should be given if this is a novel mAb. If it is the same mAb as published by Chi et al., this should be more clearly stated.

Response: We apologize, as this was not the same mAb but was produced similarly. The former mAb was not found to neutralize IgA1P activity and thus, we proceeded to test a second mAb reported here that did neutralize IgA1P activity. We have referred to the first non-neutralizing mAb as mAb #1 and the newly discovered neutralizing mAb as mAb #2 for clarity.

Line 267-269: Ying-Chih Chi is not included in the author contributions.

Response: We have now included the contribution.

Fig. 2d: It is not clear what the dashed box is referring to. It does not correspond to the expansion shown in Fig. 2e.

Response: This dashed box was to indicate the blowup for Fig. 2e that was also rotated 45 degrees. We have removed this dashed box as not to confuse the reader and apologize for any lack of clarity.

Fig. 4b, e: HC-mAb is mistakenly labelled as Fc-mAb contradicting the figure text. Labelling the blue as VH-CH1 and the yellow as VL-CL would be preferable.

Response: This has been changed accordingly within both the figure and the manuscript.

Supplementary data, line 98: How was neutralization percentage calculated from this two measurements? This is not clear from the text.

Response: We have rewritten the neutralization assay that includes the explicit calculation of the percent neutralization.

Fig. S1d: Y-axes labelling as “intensity” is unclear. Is it absorption at 280 nm?

Response: Yes, we have changed this explicitly to “Absorbance (280 nm)”.

Fig. S5a: Concentration of IgA1P should be mentioned and what the molar ratios between IgA1P and the mAb are. Otherwise the potency of the mAb cannot be judged. It is moreover unclear what the error bars show.

Response: We apologize for this lack of information and have now explained this more in depth. This is now Figure 4a. Error bars represent the average from four repeats of this ELISA-based neutralization assays that each comprised two experimental duplicates. The explicit molar concentrations are also stated within the figure legend so that the reader can quickly note the molar ratios as well.

Fig. S5b: Scale is missing.

Response: We have corrected this.

REVIEWERS' COMMENTS

Reviewer #1 (Remarks to the Author):

The authors have more or less addressed most of my previous concerns.

Reviewer #3 (Remarks to the Author):

The authors have addressed all my concerns adequately and made the requested changes to the manuscript.